# Debiasing Scores and Prompts of 2D Diffusion for View-consistent Text-to-3D Generation

**Susung Hong**[*]  **Donghoon Ahn**[*]  **Seungryong Kim**[†]

Korea University, Seoul, Korea

## Abstract

Existing score-distilling text-to-3D generation techniques, despite their considerable promise, often encounter the view inconsistency problem. One of the most notable issues is the Janus problem, where the most canonical view of an object (*e.g.*, face or head) appears in other views. In this work, we explore existing frameworks for score-distilling text-to-3D generation and identify the main causes of the view inconsistency problem—the embedded bias of 2D diffusion models. Based on these findings, we propose two approaches to debias the score-distillation frameworks for view-consistent text-to-3D generation. Our first approach, called score debiasing, involves cutting off the score estimated by 2D diffusion models and gradually increasing the truncation value throughout the optimization process. Our second approach, called prompt debiasing, identifies conflicting words between user prompts and view prompts using a language model, and adjusts the discrepancy between view prompts and the viewing direction of an object. Our experimental results show that our methods improve the realism of the generated 3D objects by significantly reducing artifacts and achieve a good trade-off between faithfulness to the 2D diffusion models and 3D consistency with little overhead. Our project page is available at `https://susunghong.github.io/Debiased-Score-Distillation-Sampling/`.

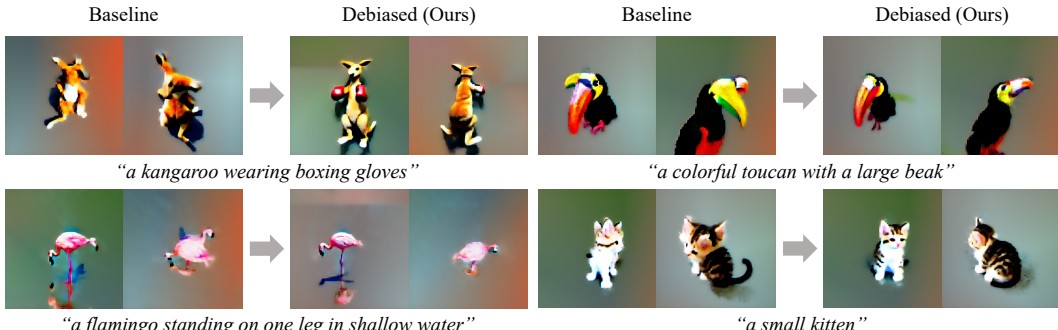

Baseline — Debiased (Ours) — Baseline — Debiased (Ours)

*"a kangaroo wearing boxing gloves"*      *"a colorful toucan with a large beak"*

*"a flamingo standing on one leg in shallow water"*      *"a small kitten"*

Figure 1: **Comparison between the baseline (SJC [27]) and ours (Debiased Score Distillation Sampling; D-SDS).** Our debiasing methods qualitatively reduce view inconsistencies in zero-shot text-to-3D generation and the so-called *Janus problem*.

---

[*]Equal contribution.

[†]Corresponding author

37th Conference on Neural Information Processing Systems (NeurIPS 2023).

# 1    Introduction

Recently, significant advancements have been made in the field of zero-shot text-to-3D generation [8], particularly with the integration of score-distillation techniques [10, 27, 18, 13] and diffusion models [9, 25, 24, 5, 21, 20, 23, 2] to optimize neural radiance fields [15]. These methods provide a solution for generating a wide range of 3D objects from a textual input, without requiring 3D supervision. Despite their considerable promise, these approaches often encounter the view inconsistency problem. One of the most notable problems is the multi-face issue, also referred to as *the Janus problem* [18], which is illustrated in the "Baseline" of Fig. 1. This problem constrains the applicability of the methods [10, 27, 18, 13], but the Janus problem is rarely formulated or carefully analyzed in previous literature.

To address the problem of view inconsistency, we delve into the formulation of score-distilling text-to-3D generation presented in [18, 27, 13, 10]. We generalize and expand upon the assumptions about the gradients concerning the parameters of a 3D scene in previous works such as DreamFusion [18] and Score Jacobian Chaining (SJC) [27], and identify the main causes of the problem within the estimated score. The score can be further divided into the unconditional score and pose-prompt gradient, both of which interrupt the estimation of unbiased gradients concerning the 3D scene. Additionally, since a naive text prompt describes a canonical view of an image such as front view, prior text-to-3D generation works [22, 18, 27, 13, 10] append a view prompt (*e.g.*, "front view", "side view", "back view", "overhead view", *etc*.) to the user's input, depending on the sampled camera angle, to better reflect its appearance from a different view. We present an analysis of the score with user prompts and view prompts and their effect on 3D, arguing that refining them is necessary for generating more realistic and view-consistent 3D objects.

Building on this concept and drawing inspiration from gradient clipping [14] and dynamic thresholding [21], we propose a score debiasing method that performs dynamic score clipping. Specifically, our method cuts off the score estimated by 2D diffusion models to mitigate the impact of erroneous bias (Fig. 2 and Fig. 3). With this debiasing approach, we reduce artifacts in the generated 3D objects and alleviate the view inconsistency problem by striking a balance between faithfulness to 2D models and 3D consistency. Furthermore, by gradually increasing the truncation value, which aligns with the coarse-to-fine nature of generating 3D objects [3, 15], we achieve a better trade-off for 3D consistency without significantly compromising faithfulness.

While the first attempt to address the bias issue in the scores, we further present a prompt debiasing method. In contrast to prior works [18, 27, 10] that simply concatenate a view prompt and user prompt, our method reduces inherent contradiction between them by leveraging a language model trained with a masked language modeling (MLM) objective [1], computing the point-wise mutual information. Additionally, we decrease the discrepancy between the assignment of the view prompt and camera pose by adjusting the range of view prompts. These enable text-to-image models [21, 20, 16] to predict accurate 2D scores, resulting in 3D objects that possess more realistic and consistent structures.

# 2    Background

**Diffusion models.**    Denoising diffusion models [5, 23] generate images through progressive denoising process. During training, denoising diffusion probabilistic models (DDPM) [5] optimizes the following simplified objective:

$$L_{\text{DDPM}} := \mathbb{E}_{\boldsymbol{\epsilon} \sim \mathcal{N}(0,\mathbf{I}), \mathbf{x}_0, t} \left[ \left\| \boldsymbol{\epsilon} - \boldsymbol{\epsilon}_\phi(\mathbf{x}_t, t) \right\|^2 \right], \tag{1}$$

where $\boldsymbol{\epsilon}_\phi$ is a network of the diffusion model parameterized by $\phi$, $t \in \{T, T-1, ..., 1\}$ is a timestep, $\mathbf{x}_0$ is an original image, and $\mathbf{x}_t$ denotes a perturbed image according to the timestep $t$. During inference, starting from $\mathbf{x}_t$, DDPM samples a previous sample $\mathbf{x}_{t-1}$ from a normal distribution with probability density of $p_\phi(\mathbf{x}_{t-1}|\mathbf{x}_t)$.

Some works fit DDPM into the generalized frameworks, *e.g.*, non-Markovian [23], score-based [25, 9], *etc*. Notably, denoising diffusion models have a tight relationship with score-based models [25, 9] in the continuous form. Furthermore, it has been shown that denoising diffusion models can be refactored into the canonical form of denoising score matching using the same network parameterization [9].

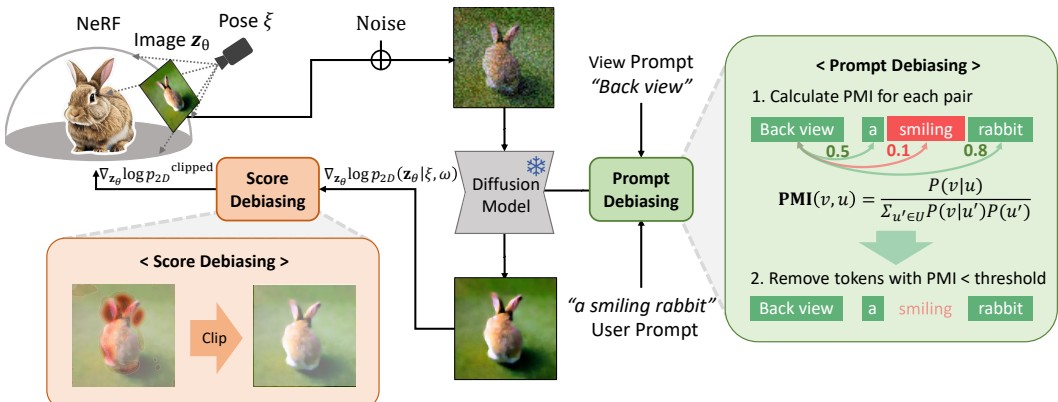

Figure 2: **Illustration of our framework.** We propose prompt and score debiasing techniques to estimate robust and unbiased gradients of the 3D parameters w.r.t. the viewpoints.

This formulation further facilitates the direct computation of 2D scores [24, 9] with the following equation:

$$\nabla_{\mathbf{x}} \log p(\mathbf{x}; \sigma) = \frac{D_\phi(\mathbf{x}; \sigma) - \mathbf{x}}{\sigma^2}, \tag{2}$$

where $D_\phi$ is an optimal denoiser network trained for every $\sigma$. With some preconditioning, a diffusion model $\boldsymbol{\epsilon}_\phi$ [5, 23, 17, 20] turns into a denoiser $D_\phi$.

Recent advancements in diffusion models have sparked increased interest in text-to-image generation [19–21, 16]. Diffusion guidance techniques [2, 6, 16, 7] have been developed to enable the control of the generation process based on various conditions such as class labels [6, 2], text captions [16], or internal information [7]. In particular, our work conditions text prompts with classifier-free guidance [6], which is formulated as follows given a conditional diffusion model $\boldsymbol{\epsilon}_\phi(\mathbf{x}_t, t, \omega)$:

$$\tilde{\boldsymbol{\epsilon}} = \boldsymbol{\epsilon}_\phi(\mathbf{x}_t, t, \omega) + s \cdot (\boldsymbol{\epsilon}_\phi(\mathbf{x}_t, t, \omega) - \boldsymbol{\epsilon}_\phi(\mathbf{x}_t, t)), \tag{3}$$

where $\tilde{\boldsymbol{\epsilon}}$ is the guided output, $\omega$ is the user-given text prompt (*user prompt* for brevity), and $s$ is the guidance scale [6].

**Score distillation for text-to-3D generation.** Diffusion models have shown remarkable performance in text-to-image modeling [16, 19, 21, 7, 20]. On top of this, DreamFusion [18] proposes the score-distillation sampling (SDS) method that uses text-to-image diffusion models to optimize neural fields [15], achieving encouraging results. The score-distillation sampling utilizes the gradient computed by the following equation:

$$\nabla_\theta L_{\text{SDS}} \triangleq \mathbb{E}_{\boldsymbol{\epsilon} \sim \mathcal{N}(0,\mathbf{I}),t} \left[ w(t)(\boldsymbol{\epsilon}_\phi(\mathbf{z}_t, t, \omega) - \boldsymbol{\epsilon}) \frac{\partial \mathbf{z}_\theta}{\partial \theta} \right], \tag{4}$$

where $\mathbf{z}_t$ denotes the $t$-step noised version of $\mathbf{z}_\theta$ which is a rendered image from a NeRF network with parameters $\theta$ [15], and $w(t)$ is a scaling function only dependent on $t$. This gradient omits the Jacobian of the diffusion backbone, leading to tractable optimization in differentiable parameterizations [18].

On the other hand, in light of the interpretation of diffusion models as denoisers, SJC [27] presents a new approach directly using the score estimation, called perturb-and-average scoring (PAAS). The work shows that the U-Net Jacobian emerging in DreamFusion is not even necessary, as well as forming a strong baseline using publicly open Stable Diffusion [20]. The perturb-and-average score approximates to a score with an inflated noise level:

$$\nabla_{\mathbf{z}_\theta} \log p_{\sqrt{2}\sigma}(\mathbf{z}_\theta) \approx \mathbb{E}_{n \sim \mathcal{N}(0,\mathbf{I})} \left[ \frac{D_\theta(\mathbf{z}_\theta + \sigma n; \sigma) - \mathbf{z}_\theta}{\sigma^2} \right], \tag{5}$$

where the expectation is practically estimated by Monte Carlo sampling. This score estimate is then directly plugged into the 2D-to-3D chain rule and produces:

$$\nabla_\theta L_{\text{PAAS}} \triangleq \mathbb{E}_{\mathbf{z}_\theta} \left[ \nabla_{\mathbf{z}_\theta} \log p_{\sqrt{2}\sigma}(\mathbf{z}_\theta) \frac{\partial \mathbf{z}_\theta}{\partial \theta} \right]. \tag{6}$$

Although the derivation is different from SDS in DreamFusion [18], it is straightforward to show that the estimation $\nabla_\theta L_{\text{PAAS}}$ is the same as $\nabla_\theta L_{\text{SDS}}$ with a different weighting rule and sampler [9].

In general, frameworks distilling the score of text-to-image diffusion models [18, 10, 27, 13, 11, 22] achieve a certain level of view consistency by concatenating view prompts (*e.g.*, "back view of") with user prompts [18, 10, 27, 13, 11, 22]. Although this is an important part in score distillation, it is rarely discussed. In the following section, we elucidate this altogether, uncovering the underlying causes of the Janus problem.

## 3 Score Distillation and the Janus Problem

SJC [27] defines the probability density function of parameters $\theta$ of 3D volume (*e.g.*, NeRF [15]) as an expectation of the likelihood of 2D rendered images $\mathbf{z}_\theta$ from uniformly sampled object-space viewpoints (Eq. 6 in [27]). Unlike this definition, our approach defines the density function of the parameters $\theta$ as a product of conditional likelihoods given a set of uniformly sampled viewpoints $\Lambda$ and user prompt $\omega$. This can be expressed as:

$$\tilde{p}_{\text{3D}}(\theta) = \prod_{\lambda \in \Lambda} p_{\text{2D}}(\mathbf{z}_\theta | \lambda, \omega), \tag{7}$$

where $p_{\text{2D}}$ and $\tilde{p}_{\text{3D}}$ denote the probability density of 2D image distribution and unnormalized density of 3D parametrizations, respectively. By using this formulation, we avoid using Jensen's inequality, in contrast to [27]. Applying the logarithm to each side of the equation yields:

$$\log \tilde{p}_{\text{3D}}(\theta) = \sum_{\lambda \in \Lambda} \log p_{\text{2D}}(\mathbf{z}_\theta | \lambda, \omega). \tag{8}$$

By taking the gradient of $\log \tilde{p}_{\text{3D}}(\theta)$, we can directly obtain $\nabla_\theta \log p_{\text{3D}}(\theta)$, since the normalizing constant of $\tilde{p}_{\text{3D}}$ is irrelevant to $\theta$. Using the chain rule, we obtain:

$$\nabla_\theta \log p_{\text{3D}}(\theta) = \sum_{\lambda \in \Lambda} \nabla_\theta \log p_{\text{2D}}(\mathbf{z}_\theta | \lambda, \omega) = Z \cdot \mathbb{E}_{\lambda \in \Lambda} \left[ \nabla_\theta \log p_{\text{2D}}(\mathbf{z}_\theta | \lambda, \omega) \right]$$
$$= Z \cdot \mathbb{E}_{\lambda \in \Lambda} \left[ \nabla_{\mathbf{z}_\theta} \log p_{\text{2D}}(\mathbf{z}_\theta | \lambda, \omega) \frac{\partial \mathbf{z}_\theta}{\partial \theta} \right], \tag{9}$$

where $Z = |\Lambda|$ is a constant, and $\nabla_{\mathbf{z}_\theta} \log p_{\text{2D}}(\mathbf{z}_\theta | \lambda, \omega)$ is practically estimated by diffusion models [9]. Note that this definition generalizes SJC [27] and even $\nabla_\theta \mathcal{L}_{\text{SDS}}$ in DreamFusion[18], which can be easily seen as the estimation of Eq. 9 with a different weighting rule and sampler. This is further expanded by applying Bayes' rule as follows:

$$\nabla_\theta \log p_{\text{3D}}(\theta) = Z \cdot \mathbb{E}_{\lambda \in \Lambda} \left[ \Big( \underbrace{\nabla_{\mathbf{z}_\theta} \log p_{\text{2D}}(\mathbf{z}_\theta)}_{\text{Unconditional score}} + \underbrace{\nabla_{\mathbf{z}_\theta} \log p_{\text{2D}}(\lambda, \omega | \mathbf{z}_\theta)}_{\text{Pose-prompt gradient}} \Big) \frac{\partial \mathbf{z}_\theta}{\partial \theta} \right]. \tag{10}$$

The first gradient term, reflecting the unconditional score modeled by 2D diffusion models [5, 25], contains a bias that affects images viewed from typical viewpoints during early optimization of 3D volume when $\mathbf{z}_\theta$ is noisy. This contributes to the Janus problem, as facial views are more prevalent in the 2D data distribution for some objects.

On the other hand, the pose-prompt gradient in Eq. 10 is guidance [25, 6, 2, 7] that drives the rendered image to better represent a specific camera pose and user prompt. The term is further expanded:

$$\nabla_{\mathbf{z}_\theta} \log p_{\text{2D}}(\lambda, \omega | \mathbf{z}_\theta) = \nabla_{\mathbf{z}_\theta} \log p_{\text{2D}}(\lambda | \mathbf{z}_\theta) + \nabla_{\mathbf{z}_\theta} \log p_{\text{2D}}(\omega | \mathbf{z}_\theta) + \nabla_{\mathbf{z}_\theta} \log C, \tag{11}$$

where $C$ is defined as $\frac{p_{\text{2D}}(\lambda, \omega | \mathbf{z}_\theta)}{p_{\text{2D}}(\lambda | \mathbf{z}_\theta) p_{\text{2D}}(\omega | \mathbf{z}_\theta)} = \frac{p_{\text{2D}}(\lambda | \omega, \mathbf{z}_\theta)}{p_{\text{2D}}(\lambda | \mathbf{z}_\theta)}$, which represents the pointwise conditional mutual information (PCMI). If a viewpoint $\lambda$ and a user prompt $\omega$ are contradictory, *i.e.*, $p_{\text{2D}}(\lambda | \omega, \mathbf{z}_\theta) \ll p_{\text{2D}}(\lambda | \mathbf{z}_\theta)$, then $C$ approximates to 0 for every $\mathbf{z}_\theta$. Simultaneously, the terms $\nabla_{\mathbf{z}_\theta} \log p_{\text{2D}}(\lambda | \mathbf{z}_\theta)$ and $\nabla_{\mathbf{z}_\theta} \log p_{\text{2D}}(\omega | \mathbf{z}_\theta)$ have an adverse effect on the 3D scene, making the view-consistent optimization particularly challenging.

## 4 Methodology

### 4.1 Score debiasing

**Motivation and overview.** If the unconditional score, $\nabla_{\mathbf{z}_\theta} \log p_{2D}(\mathbf{z}_\theta)$, is biased towards some viewing directions, which is likely in 2D data as mentioned in Sec. 3, it can negatively affect the 3D consistency and realism of generated objects through the chain rule (Eq. 9). Moreover, large magnitudes in the user prompt gradient, $\nabla_{\mathbf{z}_\theta} \log p_{2D}(\omega|\mathbf{z}_\theta)$, can also cause issues by introducing text-related artifacts that are not present in the image rendered from a 3D field (see Fig. 1 and Fig. 3). Such artifacts include extra faces, beaks, and horns, which are unrealistic or inconsistent with the 3D object's structure.

High magnitude in those two terms is typically observed when the perturbed-and-denoised image by diffusion models significantly deviates from the rendered image in the corresponding pixels (Fig. 3). Hence, adjusting this gradient is necessary to reduce the artifacts and improve the realism of the generated 3D objects. However, the 2D bias that flows into the 3D field has hardly been formulated or adjusted for better optimization and 3D consistency.

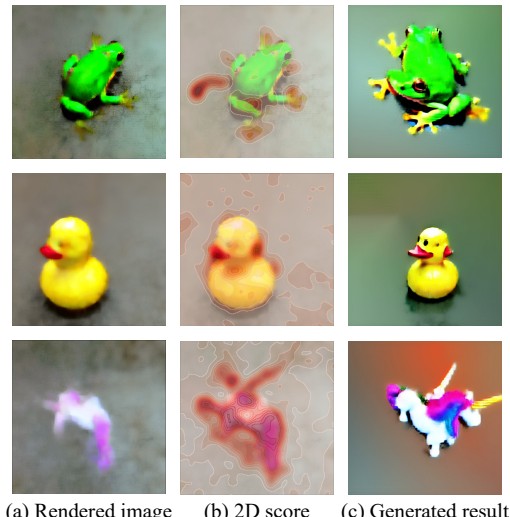

The intuition behind the scale of the distilled score $\nabla_{\mathbf{z}_\theta} \log p_{\sqrt{2}\sigma}(\mathbf{z}_\theta)$ can be mathematically elucidated by examining its relationship with the expectation term. Concretely, the distilled score serves as an approximation of the expected value of the difference between the distorted image $D(\mathbf{z}_\theta + \sigma n; \sigma)$ and the original rendered image $\mathbf{z}_\theta$, normalized by the square of the noise scale. This expectation is evaluated with respect to the normal distribution $\mathcal{N}(0, \mathbf{I})$ from which the noise term $n$ is sampled. Note that the noise term not only facilitates the use of diffusion models but can also be interpreted as a random perturbation applied to the rendered image $\mathbf{z}_\theta$.

(a) Rendered image    (b) 2D score    (c) Generated result

Figure 3: **Visualization of the magnitude of the estimated $\nabla_{\mathbf{z}_\theta} \log p_{2D}(\mathbf{z}_\theta|\lambda, \omega)$ during the optimization.** This visualization demonstrates that erroneous 2D scores result in critical artifacts, *e.g.*, additional legs, beaks, and horns in this figure.

In this context, the expectation term provides a measure of the sensitivity of the denoising process to variations in the noise. In other words, the magnitude of the estimated score can be interpreted as the (scaled) deviation of the original rendered image $\mathbf{z}_\theta$ from the 3D field. Notably, NeRF-W [12] also provides a mechanism for handling uncertainty by explicitly rendering the variance. On the contrary, we propose a novel and efficient method to directly clamp the estimated score, effectively suppressing significant deviations that ignore either geometry or appearance, thereby addressing the intrinsic bias inherent in score-based models.

**Dynamic clipping of 2D-to-3D scores.** In light of the need to control the flow of 2D scores to 3D volume (Sec. 4.1) and inspired by the clipping methods [14, 21], we propose an effective method that truncates the scores to mitigate the effects of bias and artifacts in the predicted 2D scores:

$$\nabla_{\mathbf{z}_\theta} \log p_{2D}^{\text{clipped}} = \text{Clip}(\nabla_{\mathbf{z}_\theta} \log p_{2D}(\mathbf{z}_\theta|\lambda, \omega), \psi_{\text{static}}), \tag{12}$$

where $\text{Clip}(x, c) = \max(\min(x, c), -c)$. This score clipping prevents artifacts such as extra faces, horns, eyes, and ears from appearing on the 3D objects.

However, the application of naive score clipping creates a large threshold-dependent tradeoff between 3D consistency and 2D fidelity: the lower the threshold, the more artifacts are removed, but at the expense of 2D fidelity. To circumvent this, we introduce an effective coarse-to-fine strategy [15, 3]:

$$\psi_{\text{dynamic}} := (1 - \tau)\psi_{\text{start}} + \tau\psi_{\text{end}},$$
$$\nabla_{\mathbf{z}_\theta} \log p_{2D}^{\text{clipped}} = \text{Clip}(\nabla_{\mathbf{z}_\theta} \log p_{2D}(\mathbf{z}_\theta|\lambda, \omega), \psi_{\text{dynamic}}), \tag{13}$$

where $\tau = \frac{\text{(step)}}{\text{(max step)}}$. In the early stages of optimization, we focus on the overall structure and shape, which do not require the large magnitudes of the 2D scores, while in later stages, we focus more on the details that require higher magnitudes, so we increase the threshold as the optimization progresses. We provide an illustration in Appendix A.5 to show what the rendered image at each step looks like as the scene undergoes optimization.

## 4.2 Prompt debiasing

**Motivation and overview.** Text-to-3D generation methods that distill diffusion models [18, 27] achieve a certain level of view consistency by concatenating view prompts (*e.g.*, "back view of") with user prompts [18, 10, 27, 13, 11, 22]. This simple and effective method leverages the knowledge of large-scale text-to-image models.

However, we argue that the current strategy of creating a view-dependent prompt by simply concatenating a view prompt with a user prompt is intrinsically problematic, as it can result in a contradiction between them. This contradiction is one of the causes that make diffusion models not follow the view prompt.

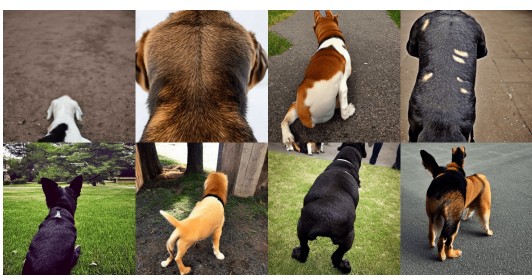

*"Back view of a dog"*

Therefore, in the following subsection, we propose identifying the contradiction between the view prompt and user prompt using off-the-shelf language models trained with masked language modeling (MLM) [1].

Additionally, instead of naively assigning regular regions for view prompt augmentations, in the next subsection, we reduce the discrepancy between the view prompt and object-space pose by adjusting the regions.

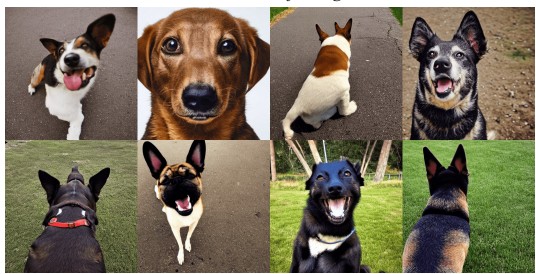

*"Back view of a smiling dog"*

**Identifying contradiction.** The prompt gradient term $\nabla_{\mathbf{z}_\theta} \log p_{2D}(\omega|\mathbf{z}_\theta)$ may cancel out the pose gradient term $\nabla_{\mathbf{z}_\theta} \log p_{2D}(\lambda|\mathbf{z}_\theta)$ needed for the view consistency of generated 3D objects, as we can derive from Eq. 11. For example, if the view prompt is "back view of" and the user prompt is "a smiling dog", it results in a contradiction since an observer cannot see the

Figure 4: **Samples from Stable Diffusion [20] given a text prompt with contradiction.** Despite "Back view of" is given in the prompts, the word "smiling" in the prompt makes diffusion models biased towards the front view of objects.

dog's smile viewing from the back. This causes diffusion models not to follow a view prompt, but instead to follow a word like "smiling" in a user prompt, as shown in Fig. 4.

In this regard, we propose a method for identifying contradictions using language models trained with masked language modeling (MLM). Specifically, let $V$ represent a set of possible view prompts, and let $U$ be a set of size 2, which contains the presence and absence of a word in the user prompt for brevity. We then compute the following:

$$\frac{P(v|u)}{P(v)} = \frac{P(v|u)}{\sum_{u' \in U} P(v|u')P(u')}, \tag{14}$$

where we technically model $P(v|u)$ with masked language modeling by alternating the view prompts and normalizing them, and $P(u)$ is a user-defined faithfulness. Note that Eq. 14 corresponds to the pointwise mutual information (PMI), as $\text{PMI}(v, u) \triangleq \frac{P(v,u)}{P(v)P(u)} = \frac{P(v|u)}{P(u)}$, and removing a contradiction involves eliminating a word with a low PMI value concerning the view prompts. In practice, a word from a user prompt is omitted if the value falls below a certain threshold.

**Reducing discrepancy between view prompts and camera poses.** Existing methods [18, 27, 10, 13] utilize view prompt augmentations by dividing the camera space into some regular sections (*e.g.,* front, back, side, and overhead in DreamFusion [18]). However, this approach does not match the real distribution of object-centric poses in image-text pairs; *e.g.,* the front view may cover a narrower

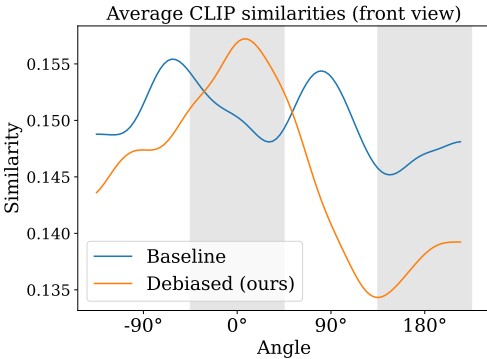 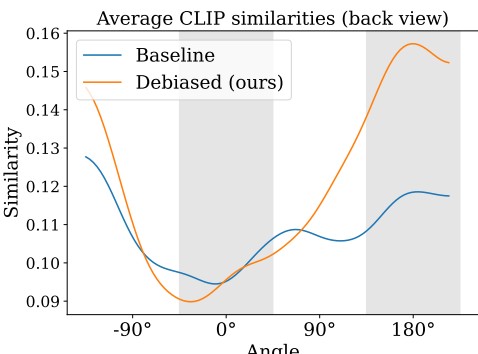

Figure 5: **Average CLIP similarities of rendered images for each azimuth, calculated using view-augmented prompts.** The shaded areas, starting from the left, represent the $90°$ regions for the front view and back view, respectively.

| Method | A-LPIPS$_{VGG}$ $\downarrow$ | A-LPIPS$_{Alex}$ $\downarrow$ |
|---|---|---|
| Baseline [27] | 0.2054 | 0.1526 |
| Debiased (Preserved) | 0.1963 | 0.1450 |
| Debiased (Ours) | **0.1940** | **0.1445** |

Table 1: **Quantitative evaluation.** The best values are in bold, and the second best are underlined. *Preserved* means user prompts are preserved, i.e., $P(u) = 1$ for all $u$.

region. Therefore, we make practical adjustments to the range of view prompts, such as reducing the azimuth range of the "front view" by half, and also search for precise view prompts [18, 27] that yield improved results.

## 5 Experiments

### 5.1 Implementation details

We build our debiasing methods on the high-performing public repository of SJC [27]. For all the results, including SJC and ours, we run 10,000 steps to optimize the 3D fields, which takes about 20 minutes using a single NVIDIA 3090 RTX GPU and adds almost no overhead compared to the baseline. We set the hyperparameters of SJC to specific constants [27] and do not change them throughout the experiments.

### 5.2 Evaluation Metrics

Quantitatively evaluating a zero-shot text-to-3D framework is challenging due to the absence of ground truth 3D scenes that correspond to the text prompts. Existing works employ CLIP R-Precision [8, 18]. However, it measures retrieval accuracy through projected 2D images and text input, making it unsuitable for quantifying the view consistency of a scene.

Therefore, to measure the view consistency of generated 3D objects quantitatively, we compute the average LPIPS [29] between adjacent images, which we refer to as A-LPIPS. We sample 100 uniformly spaced camera poses from an upper hemisphere of a fixed radius, all directed towards the sphere's center at an identical elevation, and render 100 images from a 3D scene. Then, we average the LPIPS values evaluated for all adjacent pairs of images in the 3D scene, finally aggregating those averages across the scenes. The intuition behind this is that if there exist artifacts or view inconsistencies in a generated 3D scene, the perceptual loss will be large near those points.

In addition, to assess the faithfulness to the view-augmented prompt, we present a graph that illustrates the average CLIP similarities of rendered images for each azimuth, as determined by view-augmented prompts. This metric is designed to be high when the score-distillation pipeline effectively generates an accurate view of an object.

| Stable-DreamFusion | Baseline | Debiased (Ours) |
| --- | --- | --- |

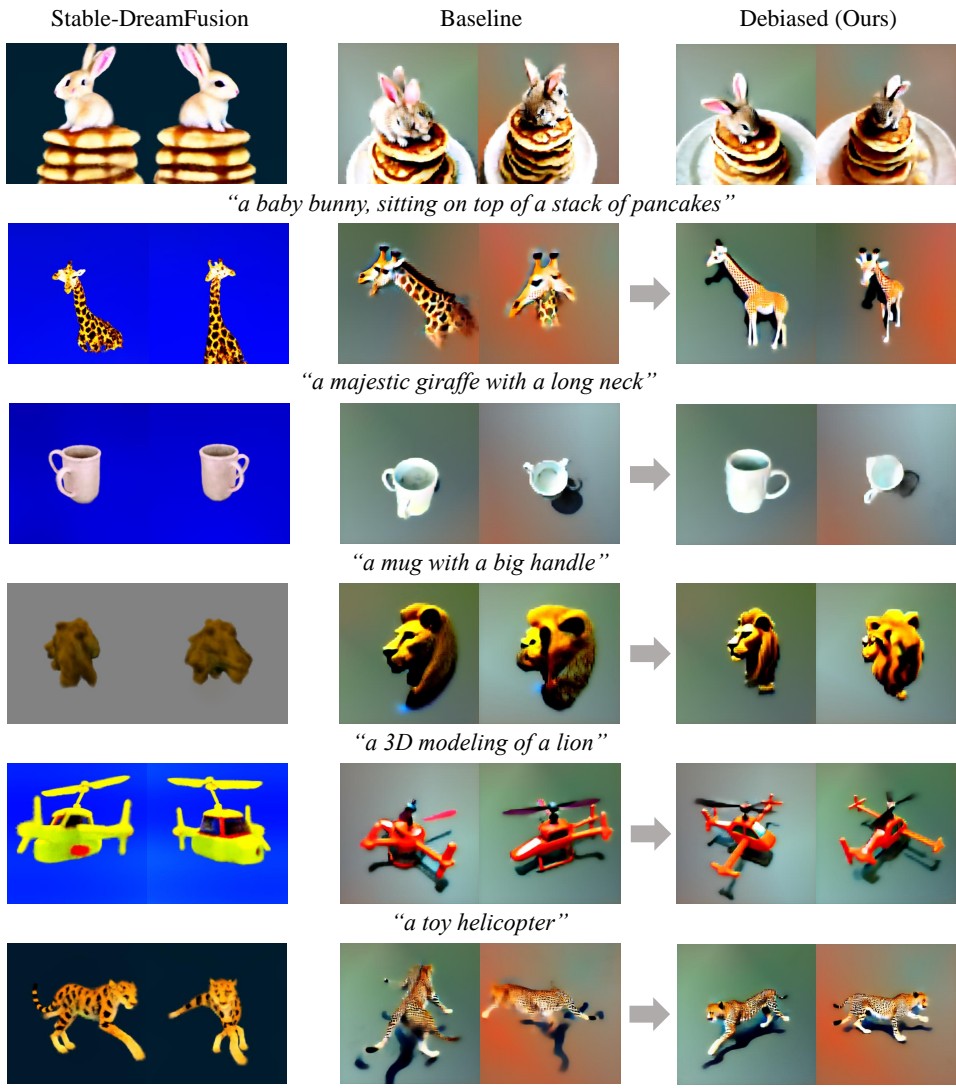

*"a baby bunny, sitting on top of a stack of pancakes"*

*"a majestic giraffe with a long neck"*

*"a mug with a big handle"*

*"a 3D modeling of a lion"*

*"a toy helicopter"*

*"a sleek and speedy cheetah in mid-stride"*

Figure 6: **Comparison between Stable-DreamFusion [26, 18], SJC [27], and ours.** The baseline is original SJC [27]. Our debiasing methods qualitatively reduce view inconsistencies in zero-shot text-to-3D and the so-called *Janus problem*.

| Method | View consistency | Faithfulness | Overall quality |
| --- | --- | --- | --- |
| Baseline [27] | 9.58% | 16.07% | 9.67% |
| Debiased (Ours) | **90.42%** | **83.93%** | **90.33%** |

Table 2: **User study.**

## 5.3 Comparison with the baseline

**Quantitative results.** We present quantitative results from 70 user prompts for the baseline [27], our combined method, and the method without the removal of contradicting words from a user prompt. Our method produces more consistent 3D objects than the baseline, as demonstrated in Table 1. Note that removing contradictions in prompts indeed leads to better results with respect to A-LPIPS, meaning that the generated objects overall in each azimuth are consistent with our debiasing methods.

We also present adherence to the view-augmented prompts in Fig. 5. The diagram illustrates shaded sections depicting the 90-degree front and back view zones, beginning from the left side. When comparing our unbiased outcomes to the baseline, we observe a clear and preferable pattern in CLIP

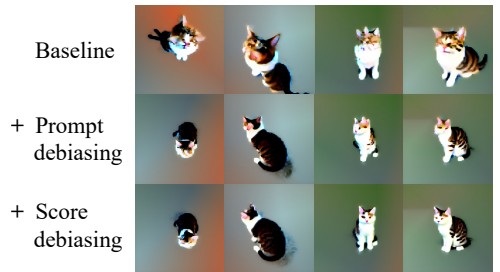 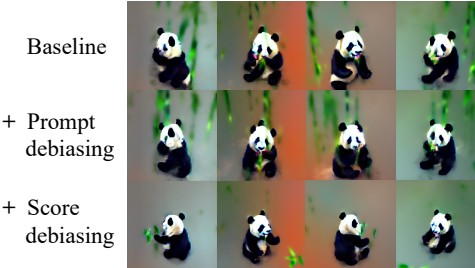

Figure 7: **Improvement of view consistency through prompt and score debiasing.** We start from the baseline (SJC [27]) and apply score debiasing and prompt debiasing sequentially for each prompt, "a smiling cat" and "a cute and chubby panda munching on bamboo", respectively.

similarities associated with the view-augmented prompts. In this pattern, the similarity with the view-augmented prompts reaches its highest point in the desired region. In contrast, the standard method exhibits minor fluctuations in CLIP similarities as we examine different angles in relation to the view prompts, implying less faithfulness to the viewing direction.

In addition to the user study outlined in Table 2, which evaluates view consistency, faithfulness to user prompts, and overall quality, we also report the success rate of the generation in Table 3. The success rate applies to 41 out of 70 prompts featuring countable faces. We marked as successful only those

| Success-Baseline | Success-Ours |
|---|---|
| 29.3% | **68.3%** |

Table 3: **Success rate.**

objects that do not exhibit the Janus problem, *i.e.*, those with an accurate number of faces. Our method significantly outperforms the baseline in terms of success rate.

Overall, the experiments corroborate that our debiasing methods improve the realism and alleviate the Janus problem of generated 3D objects, without requiring any 3D guide [22] or introducing significant overhead or additional optimization steps to the zero-shot text-to-3D setting.

**Qualitative results.** We present qualitative results in Fig. 6. In addition to the results of SJC [27], which serves as the baseline for our experiments, we include those of Stable-DreamFusion [26], an unofficial re-implementation of DreamFusion [18] that utilizes Stable Diffusion [20]. The results demonstrate that our methods significantly reduce the Janus, or view inconsistency problem. For example, given a user prompt "a majestic giraffe with a long neck," the whole body is consistently generated using our debiasing method, compared to the baseline with the Janus problem. Additionally, as a notable example, when considering "a mug with a big handle," our method successfully generates a mug with a single handle, while the counterparts generate multiple handles.

Additionally, to show that our method is not only applicable to SJC [27] with Stable Diffusion [20], but also to any text-to-3D frameworks that leverage score distillation, we present results on Dream-Fusion [18] with DeepFloyd-IF, and on concurrent frameworks such as Magic3D [10] and Prolific-Dreamer [28] in Appendix A.4, showcasing various outcomes.

## 5.4 Ablation study

**Ablation on debiasing methods.** We present ablation results in Fig. 7, where we sequentially added prompt debiasing and score debiasing on top of the baseline. This demonstrates that they gradually improve the view consistency and reduce artifacts as intended.

Using prompt debiasing alone can resolve the multi-face problem to some extent. In the case of the prompt "a smiling cat", prompt debiasing eliminates the word "smiling" from the prompt. As can be seen in column 1 and column 3, the cat has a more realistic appearance compared with the baseline. However, the cat retains an additional ear. Sometimes, such as in the instance with the panda, it can even generate a new ear. Therefore, using prompt debiasing alone does not solve the problem of creating additional artifacts like ears. Applying score debiasing removes these extra ears in both cases, leading to more view-consistent text-to-3D generation in combination with prompt debiasing.

**Ablation on dynamic clipping.** To show some examples of the effect of dynamic clipping, we compare the results with those of static clipping and no clipping in Fig. 8. It demonstrates that naive static clipping can struggle to find a good compromise between 3D consistency and 2D

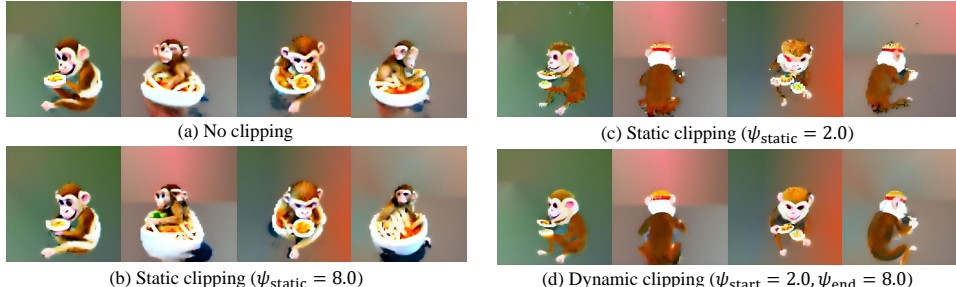

| (a) No clipping | (c) Static clipping ($\psi_{\text{static}} = 2.0$) |
| (b) Static clipping ($\psi_{\text{static}} = 8.0$) | (d) Dynamic clipping ($\psi_{\text{start}} = 2.0, \psi_{\text{end}} = 8.0$) |

Figure 8: **Dynamic clipping of 2D-to-3D scores.** The given user prompt is "a monkey eating ramen". Using static clipping, there is a tough compromise between 3D consistency and 2D faithfulness. Dynamic clipping achieves a better tradeoff between pixelation and many artifacts in the result.

faithfulness, which means lowering the threshold can eliminate more artifacts like extra ears or eyes to achieve better realism, but it also returns fairly pixelated and collapsed appearance, as can be seen in (c). Conversely, employing dynamic clipping produces visually appealing outcomes with artifacts eliminated, closely resembling the consistency of static clipping at a low threshold. Moreover, it preserves intricate shapes and details without any pixelation or degradation of the object's visual presentation.

## 6 Conclusion

In conclusion, we have addressed the critical issue of view inconsistency in zero-shot text-to-3D generation, particularly focusing on the Janus problem. By dissecting the formulation of score-distilling text-to-3D generation and pinpointing the primary causes of the problem, we have proposed a dynamic score debiasing method that mitigates the impact of erroneous bias in the estimated score. This method significantly reduces artifacts and improves the 3D consistency of generated objects. Additionally, our prompt debiasing approach refines the use of user and view prompts to create more realistic and view-consistent 3D objects. Our work, D-SDS, presents a major step forward in the development of more robust and reliable zero-shot text-to-3D generation techniques, paving the way for further advancements in the field.

## Acknowledgements

This research was supported by the MSIT, Korea (IITP-2022-2020-0-01819, ICT Creative Consilience program, RS-2023-00227592, Development of 3D Object Identification Technology Robust to Viewpoint Changes, No. 2021-0-00155, Context and Activity Analysis-based Solution for Safe Childcare), and National Research Foundation of Korea (NRF-2021R1C1C1006897).

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

# Appendix

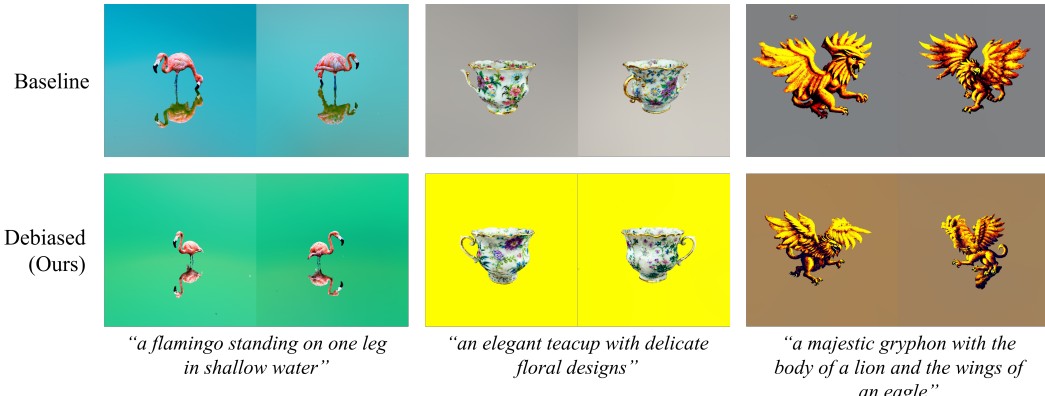

*"a flamingo standing on one leg in shallow water"*

*"an elegant teacup with delicate floral designs"*

*"a majestic gryphon with the body of a lion and the wings of an eagle"*

Figure 9: **Results of the debiased ProlificDreamer (VSD) [28] framework.** We utilize the VSD implementation, introduced in ProlificDreamer, of threestudio [4]. In the baseline examples, we observe additional necks, handles, and faces. These artifacts are mitigated in our debiased examples.

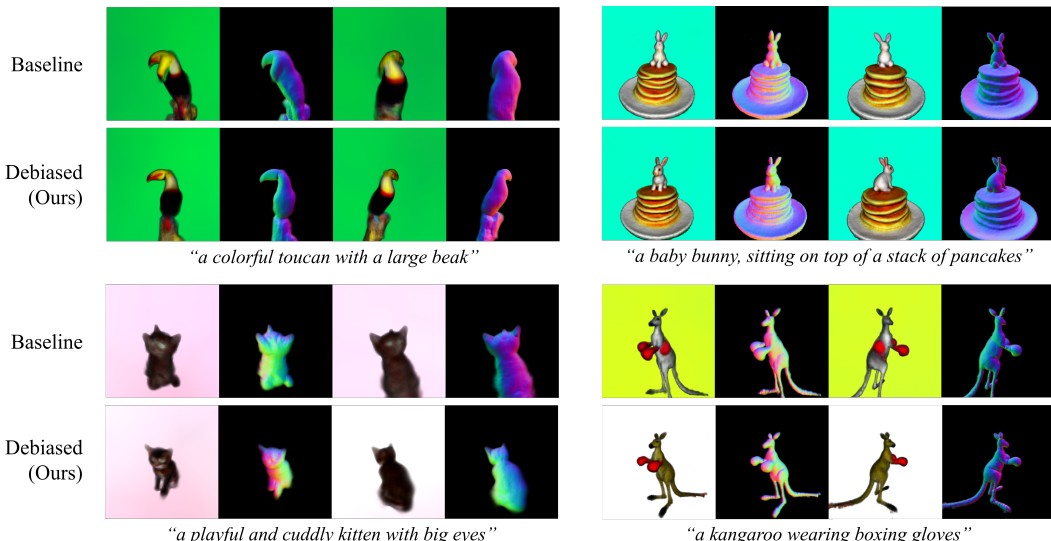

*"a colorful toucan with a large beak"*

*"a baby bunny, sitting on top of a stack of pancakes"*

*"a playful and cuddly kitten with big eyes"*

*"a kangaroo wearing boxing gloves"*

Figure 10: **Results of the debiased DreamFusion [18] framework.** We utilize the DreamFusion implementation of threestudio [4], which leverages DeepFloyd-IF.

## A   More Results

### A.1   Qualitative results

We present additional qualitative results in Figs. 12. These results clearly show that our methods alleviate the Janus problem, also known as view inconsistency.

In certain instances, even though the Janus problem is present, the images from each angle still display reasonable appearances due to smooth transitions between angles. To illustrate this, we present a series of 10 sequential images arranged in order of the camera angles, right, back, and left of the object, in Fig. 12. In the baseline images, the front view appears in the back view or side view. However, after applying our debiasing methods, we observe a significant improvement in view consistency, resulting in more realistic representations.

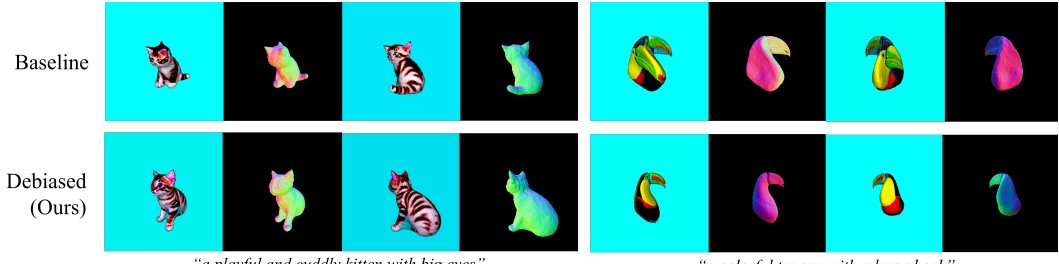

*"a playful and cuddly kitten with big eyes"*          *"a colorful toucan with a large beak"*

Figure 11: **Results of the debiased Magic3D [10] framework.** We utilize the Magic3D implementation of threestudio [4].

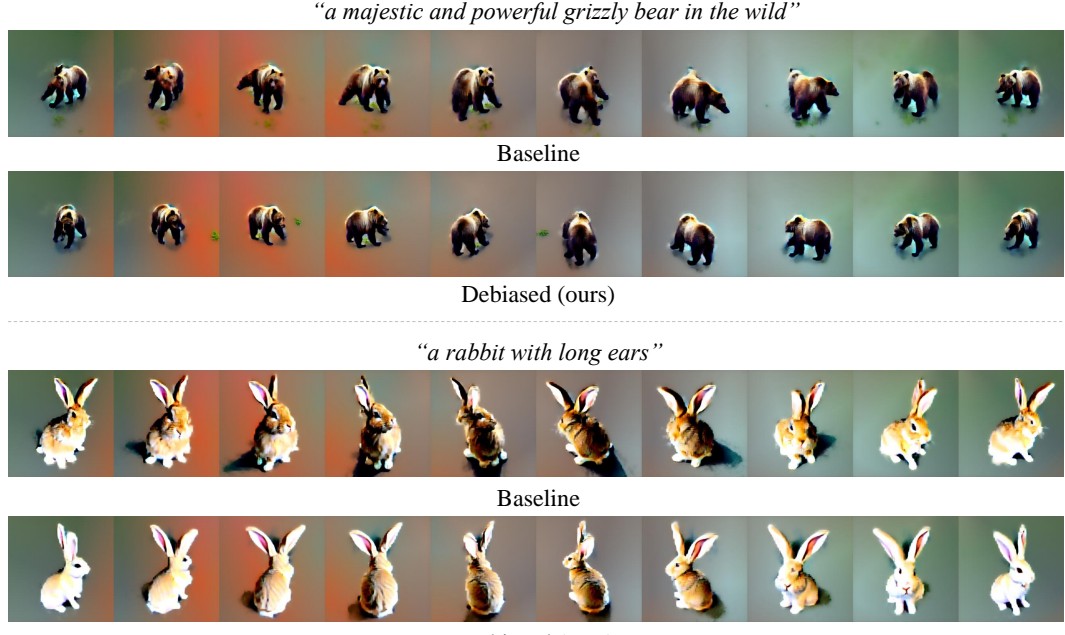

*"a majestic and powerful grizzly bear in the wild"*

Baseline

Debiased (ours)

*"a rabbit with long ears"*

Baseline

Debiased (ours)

Figure 12: **Comparison of our method with baseline (SJC [27]) in 360°.** In both cases, the baseline [27] exhibits the Janus problem, where the face appears in every view. Our debiased methods ensure proper view consistency in the 360° images.

Furthermore, we provide another example of ablation study on our methods in Fig. 13. This analysis clearly demonstrates that both prompt debiasing and score debiasing techniques significantly contribute to improved realism, reduction of artifacts, and achievement of view consistency.

## A.2 Dynamic clipping of 2D-to-3D scores

We provide an additional example where we examine the outcomes of dynamic clipping in comparison to static clipping and the absence of clipping, as shown in Fig. 14. In the case of no clipping (row (a)), several artifacts appear in certain views. Using a high threshold for static clipping yields a similar outcome (row (b)). A low threshold successfully removes artifacts, but also makes necessary objects, like icebergs, appear transparent (row (c)). Gradually reducing the threshold from high to low preserves the main object while eliminating artifacts (row (d)). Overall, this demonstrates that dynamic clipping reduces artifacts and enhances realism.

## A.3 User study

We conducted a user study to evaluate the view-consistency, faithfulness, and overall quality of the baseline and our debiased results. The results are presented in Table 2. According to the study, our

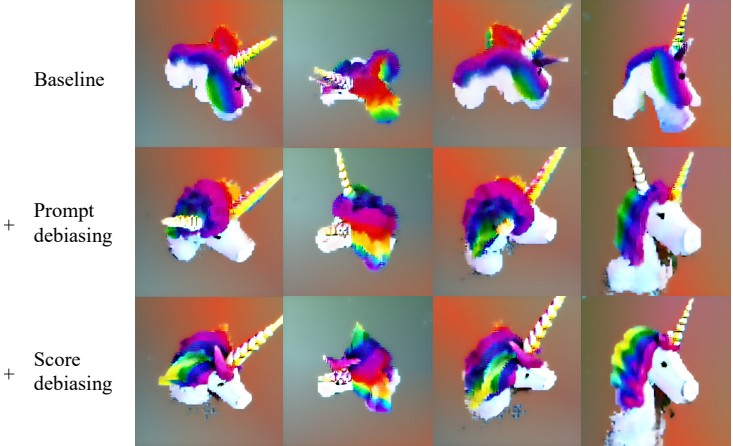

Figure 13: **Improvement of view consistency through prompt and score debiasing.** The baseline is original SJC [27], and *Prompt* and *Score* denote prompt and score debiasing, respectively. The given user prompt is *"an unicorn with a rainbow horn."*

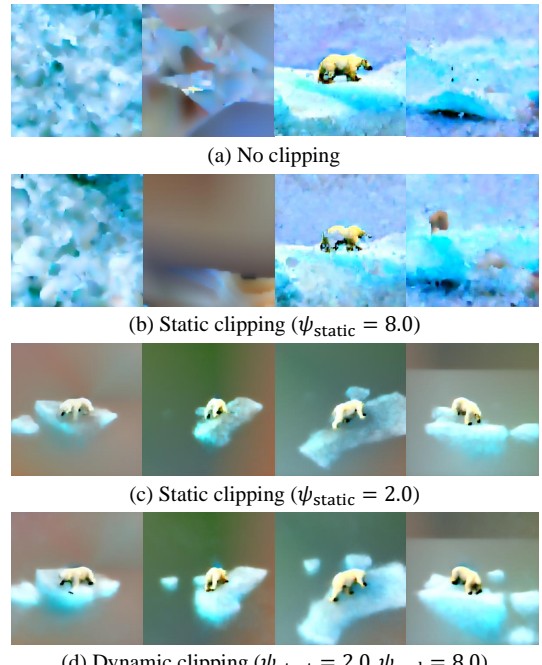

(a) No clipping

(b) Static clipping ($\psi_{\text{static}} = 8.0$)

(c) Static clipping ($\psi_{\text{static}} = 2.0$)

(d) Dynamic clipping ($\psi_{\text{start}} = 2.0, \psi_{\text{end}} = 8.0$)

Figure 14: **Dynamic clipping of 2D-to-3D scores.** The given user prompt is *"a polar bear on an iceberg".*

method surpassed the baseline in all human evaluation criteria. We tested 75 participants anonymously, and the format of instructions provided to the users was as follows:

1. Which one has a more realistic 3D form? (above/below)

2. Which one is more consistent with the prompt? Prompt: {prompt} (above/below)

3. Which one has better overall quality? (above/below)

### A.4 Results on other text-to-3D frameworks

Our method is designed to enhance 2D score-based text-to-3D generation methods. While it has been mainly claimed to be applicable to the SJC [27] and DreamFusion [18] frameworks, the applicability of our approach extends beyond these models. This approach can be adapted for any text-to-3D

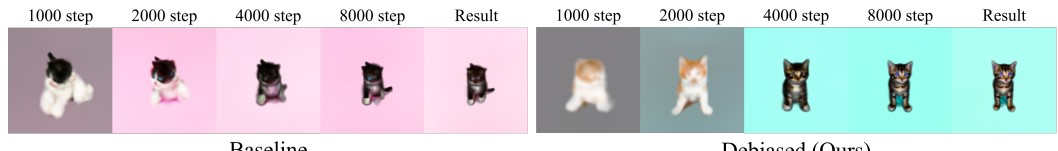

Figure 15: **Evaluation of rendered images during optimization.** Note that the geometry of the object is mostly formed within the first 4,000 optimization steps, which is when the problem in the geometry is clearly identified.

generation method that relies on a score generated by a text-to-image diffusion model and incorporates view-augmented prompting [18, 10, 28]. These methods, including contemporary works such as Magic3D [10] and ProlificDreamer [28], are to some extent susceptible to the Janus problem. With a recent implementation of the text-to-3D frameworks, threestudio [4], we have provided results that demonstrate the applicability of our method to recent frameworks such as Magic3D (Fig. 11), DreamFusion (Fig. 10), and ProlificDreamer (Fig. 9). We use the same seed for a fair comparison and only apply score debiasing for this experiment. Notably, even in instances with complex geometries that are susceptible to challenges like the Janus problem (e.g., "a majestic griffon with a lion's body and eagle's wings" or "an elegant teacup with delicate floral patterns"), the results show clear improvement when our method is applied.

### A.5 Visualization of optimization process and convergence speed

We present Fig. 15 to demonstrate how the rendered image evolves at each step during the first stage of Magic3D [10]. This experiment underscores the motivation for dynamic clipping of 2D-to-3D scores, as the geometry is determined in the early stages.

In addition, the 3D scenes for both the baseline and ours evolve similarly in terms of optimization steps, with ours being debiased. It indeed shows that the impact of gradient clipping on convergence speed is quite marginal; the number of optimization steps is comparable to that of the baseline models, and the convergence speed is nearly unchanged by our approach (approximately 20 minutes for both SJC and ours).

## B   Limitations and Broader Impact

### B.1   Limitations

Although our debiasing methods effectively tackle the Janus problem, the results produced by some prompts remain less than perfect. This is primarily due to the Stable Diffusion's limited comprehension of view-conditioned prompts. Despite the application of our debiasing methods, these inherent limitations result in constrained outputs for specific user prompts. Fig. 16 presents examples of such failure cases.

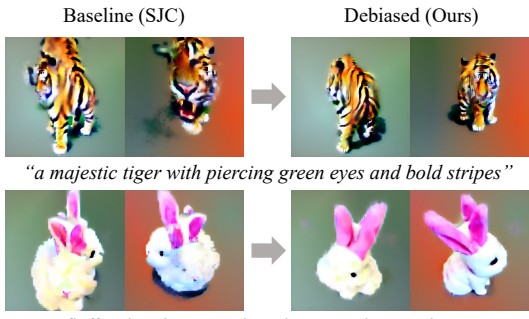

Figure 16: **Failure cases.** In some prompts where Stable Diffusion has a severely limited ability to generate view-conditioned images, the view consistency of the result is constrained.

### B.2 Broader impact

Our strategy pioneers the realm of debiasing. It possesses the capability to be integrated into any Score Distillation Sampling (SDS) technique currently under development, given that these methods universally deploy view prompts [18, 10, 27, 13, 11, 22].

Artificial Intelligence Generated Content (AIGC) has paved the way for numerous opportunities while simultaneously casting certain negative implications. However, it is important to note that our procedure is not identified as having any deleterious impact since it is exclusively designed for the purpose of debiasing the existing framework.

## C  Implementation Details

### C.1  Common settings

We base our debiasing techniques on the publicly available repository of SJC [27]. To ensure consistency, we conduct 10,000 optimization steps for both SJC and our methods to enhance the 3D fields. The hyperparameters of SJC are set to fixed values and remain unchanged throughout our experiments. For future research, we present the prompts we used in our experiments in Table 4, where some prompts are taken from DreamFusion [18], Magic3D and SJC [27]. When comparing the use of these prompts, we intentionally omit Stable Dreamfusion [26, 18] because its occasional tendency to fail to generate an object and only generate backgrounds can significantly skew our evaluation metrics.

### C.2  Score debiasing

In terms of score debiasing, we gradually increase the truncation threshold from one-fourth of the pre-defined threshold to the pre-defined threshold, according to the optimization step. Specifically, we linearly increase the threshold from $2.0$ to $8.0$ for all experiments using Stable Diffusion [20] that leverage dynamic clipping of 2D-to-3D scores. When using Deepfloyd-IF, we adopt a linearly increasing schedule from $0.5$ to $2.0$, considering the lower scale of the scores.

### C.3  Prompt debiasing

To compute the pointwise mutual information (PMI), we use the uncased model of BERT [1] to obtain the conditional probability. Additionally, we set $P(u) = 1$ for words that should not be erroneously omitted. Otherwise, we set $P(u) = 1/2$. To use a general language model for the image-related task, we concatenated "This image is depicting a" when evaluating the PMI between the view prompt and user prompt. We first get $u, v$ pairs such that $\frac{P(v,u)}{P(v)P(u)} < 1$. Then, given a view prompt, we remove words whose PMI for that view prompt, normalized across all view prompts, is below $0.95$.

For the view prompt augmentation, we typically follow the view prompt assignment rule of DreamFusion [18] and SJC [27]. However, we slightly modify the view prompts and azimuth ranges for each prompt as mentioned in Sec. 4.2. For example, we assign an azimuth range of $[-22.5°, 22.5°]$ for the "front view." Also, we empirically find that using a view prompt augmentation $v \in \{\text{"}front\ view\text{"}, \text{"}back\ view\text{"}, \text{"}side\ view\text{"}, \text{"}top\ view\text{"}\}$ without "$of$" depending on a viewpoint gives us improved results for Stable Diffusion v1.5 [20].

| |
|---|
| French fries |
| Mcdonald's fries |
| a 3D modeling of a lion |
| a baby bunny, sitting on top of a stack of pancakes |
| a blue bird with a hard beak |
| a cherry |
| a clam with pearl |
| a collie dog, high resolution |
| a crocodile |
| a dolphin swimming in the ocean |
| a dragon with a long tail |
| a flamingo standing on one leg in shallow water |
| a frog wearing a sweater |
| a kangaroo wearing boxing gloves |
| a monkey swinging through the trees |
| a mug with a big handle |
| a penguin |
| a piece of strawberry cake |
| a polar bear |
| a polar bear on an iceberg |
| a rabbit with long ears |
| a beautiful mermaid with shimmering scales and flowing hair |
| a brightly colored tree frog |
| a colorful parrot with a curved beak and vibrant feathers |
| a colorful toucan with a large beak |
| a colorful and exotic parrot with a curved beak |
| a colorful and vibrant chameleon |
| a colorful and vibrant poison dart frog perched on a leaf |
| a confident businesswoman in a sharp suit with briefcase in hand |
| a cozy coffee mug with steam rising from the top |
| a creepy spider with long, spindly legs |
| a curious meerkat standing on its hind legs |
| a curious and mischievous raccoon with a striped tail |
| a cute and chubby panda munching on bamboo |
| a cute and cuddly sugar glider jumping from tree to tree |
| a delicious sushi on a plate |
| a fearsome dragon with iridescent scales |
| a fluffy white bunny with pink ears and a twitchy nose |
| a graceful ballerina mid-dance, with flowing tutu and pointed toes |
| a majestic eagle |
| a majestic elephant with big ears and a long trunk |
| a majestic giraffe with a long neck |
| a majestic gryphon with the body of a lion and the wings of an eagle |
| a majestic lioness |
| a majestic tiger with piercing green eyes and bold stripes |
| a majestic and powerful grizzly bear in the wild |
| a photo of a comfortable bed |
| a pirate collie dog, high resolution |
| a playful baby elephant spraying water with its trunk |
| a playful dolphin leaping out of the water |
| a playful and cuddly kitten with big eyes |
| a quirky and mischievous raccoon with a striped tail |
| a refreshing and tangy grapefruit with juicy segments |
| a regal queen in a crown and elegant gown |
| a relaxed yoga practitioner in a serene pose |
| a ripe strawberry |
| a sleek and graceful shark swimming in the deep sea |
| a sleek and speedy cheetah in mid-stride |
| a sleek and speedy cheetah racing across the savannah |
| a sleek and speedy falcon soaring through the sky |
| a sleek and stealthy panther |
| a small kitten |
| a smiling cat |
| a sneaky fox |
| a toy helicopter |
| a toy sports car |
| an octopus in the ocean |
| an unicorn with a rainbow horn |
| an bitten apple |
| an elegant teacup with delicate floral designs |

Table 4: **Example prompts.**

