# OpenReview forum: "Debiasing Scores and Prompts of 2D Diffusion for View-consistent Text-to-3D Generation"
_NeurIPS.cc/2023/Conference — NeurIPS 2023 poster_

### Official Review · Reviewer_nz3q · 2023-06-10

**Soundness:** 3 good
**Presentation:** 3 good
**Contribution:** 2 fair
**Rating:** 5
**Confidence:** 5

**Summary:**

The paper proposes two simple methods to solve the widely known Janus problem in zero-shot text-to-3D generation, which is an essential issue. The proposed methods are intuitive and simple. They are more like optimization tricks instead of sysmetic formulations. From the qualitative results, the improvement of the proposed methods is not obvious.

**Strengths:**

1)The motivation is persuasive because the Janus problem is very important in text-to-3D generation.
2)The paper proposes two novel strategies of debiasing the score-distillation from 2D diffusion models, solving the Janus problem widely existing in zero-shot text-to-3D generation.
3)The first strategy performs dynamic clipping of 2D-to-3D scores to eliminate the biases towards some viewing directions, which solves the problems of additional legs, beaks, and horns.
4)The second strategy uses a pre-trained LLM to identify and remove the conflict words with the view points.

**Weaknesses:**

1)Technically, the novelty is a little weak.
2)The paper proposes to debias scores and prompts of 2D text-image diffusion models. But these two debiasing methods are simply integrated for text-3D generation without elegant co-formulation.
3)Even with sophisticated mathematical formulations, the actual debiasing score method is very intuitive and simple.
4)The prompt debiasing method is also very simple and intuitive. What is the difference between the proposed method and using the CLIP similarity scores to remove conflict words?
5)The qualitative comparisons in Figure 6 and 7 do not show much superiority of the proposed method.

**Questions:**

See weakness

**Limitations:**

See weakness

---

> ### Author Rebuttal · Authors · 2023-08-08
>
> **Qualitative results.** The reviewer mentioned that the qualitative comparisons in Figs. 6 and 7 do not show a clear improvement of the proposed method. However, we would like to note that in the supplementary material, we have provided more examples (Figs. 1 and 3) and images from 360-degree angles (Fig. 2) to better demonstrate the improvements in terms of the Janus problem. If the reviewer regards these comparisons with SJC [Wang et al., 2023] as unappealing, we have also provided a PDF file in the general response. This file includes the results of integrating our method into a variety of concurrent methods that use score distillation for text-to-3D models, such as Magic3D [Lin et al., 2023] (Fig. 3), DreamFusion [Poole et al., 2023] (Fig. 2), and ProlificDreamer [Wang et al., 2023] (Fig. 1), showing clear improvements in the Janus problem. We kindly direct the reviewer to the supplementary material and the general response for more information.
>
> **Novelty and simplicity.** The reviewer has expressed concerns regarding the perceived lack of novelty in our proposed methods. However, to the best of our knowledge, ours is the pioneering effort that directly addresses the Janus problem, and it is also the first to challenge the intrinsic bias present in scores and prompts within text-to-3D frameworks. Technically, we've leveraged a language model trained with an MLM objective and seamlessly integrated it into our framework for detecting contradictions, utilizing its probability output (Section 4.2). Additionally, the dynamic clipping of 2D-to-3D scores is not only deeply rooted in a coarse-to-fine strategy (Section 4.1) but has also demonstrated its effectiveness, as validated in the main paper Fig. 8, supplementary material A.2, and general response PDF Fig. 4. We believe that simplicity in research is not a detriment. When a straightforward method addresses an issue, it stands a better chance of being integrated into diverse frameworks due to its clarity, ease of implementation, and adaptability. However, we acknowledge the reviewer's concern and sincerely appreciate the thorough review.
>
> **Intuitiveness and simple integration.** We acknowledge that our two debiasing methods might seem simple and intuitive. In fact, in Section 4, we presented our methods in such an intuitive manner to clearly convey our motivation. We would like to emphasize the value of this intuitiveness and simplicity in their explanations; they are both explainable and adaptable wherever this intuition is applicable, notably in any text-to-3D framework that utilizes scores from text-to-image diffusion models. Moreover, our methodologies aim to solve separate problems stemming from a fundamental formulation in Section 3. Specifically, as discussed in Section 4.1, the score debiasing method offers a novel interpretation of the scores as outlined in Eq. 2, conceptualizing their magnitude as a (scaled) deviation from the rendered image of a 3D field. This approach directly suppresses deviations that ignore either viewpoint or geometry, addressing the intrinsic bias of score-based models detailed in Section 3. Concurrently, prompt debiasing arises from the need to reduce the contradiction between view prompts and user prompts, as detailed in Section 3. To address this effectively, we employed an MLM-pretrained model to calculate conditional probabilities, identify contradictions, and remove them from view-augmented prompts. Apparently, addressing both problems is necessary and complementary.
>
> **Prompt debiasing with CLIP.** The reviewer questions the difference between the proposed prompt debiasing method and the use of CLIP similarity scores to remove conflicting words. However, CLIP was designed for vision-language contrastive learning, not for calculating conditional probabilities between words. Therefore, it may not be straightforward to use CLIP similarity scores for this purpose. In contrast, our proposed method indeed calculates the desired conditional probability of a word's occurrence based on the formulation of masked language modeling. In fact, in the early stages of our research, we tried using CLIP similarity between word embeddings but found it unsuccessful, leading us to develop a more novel and sound method: the current prompt debiasing.
>
> [Poole et al., 2023] DreamFusion: Text-to-3D using 2D Diffusion, ICLR 2023.
>
> [Wang et al., 2023] Score Jacobian Chaining: Lifting Pretrained 2D Diffusion Models for 3D Generation, CVPR 2023.
>
> [Lin et al., 2023] Magic3D: High-Resolution Text-to-3D Content Creation, CVPR 2023.
>
> [Wang et al., 2023] ProlificDreamer: High-Fidelity and Diverse Text-to-3D Generation with Variational Score Distillation, arXiv preprint.

---

> > ### Comment · Reviewer_nz3q · 2023-08-12
> >
> > Thank you very much for your detailed explanation of my concerns. I keep my initial score.

---

> > > ### Author Response · Authors · 2023-08-13
> > > **Response**
> > >
> > > Thank you very much for your comments! We appreciate them and will further refine our paper.

---

### Official Review · Reviewer_9gp3 · 2023-06-28

**Soundness:** 3 good
**Presentation:** 3 good
**Contribution:** 2 fair
**Rating:** 6
**Confidence:** 5

**Summary:**

This paper proposes two approaches to debias the score-distillation frameworks for view-consistent text-to-3D generation. The first approach is called score debiasing, involves cutting off the score estimated by 2D diffusion models and gradually increasing the truncation value throughout the optimization process. The second approach, called prompt debiasing, identifies conflicting words be tween user prompts and view prompts using a language model, and adjusts the discrepancy between view prompts and the viewing direction of an object. The proposed results is demonstrated by experiments.

**Strengths:**

1. This paper is clearly written.
2. The proposed method is technically sound and insightful.
3. Some experiments are conducted to demonstrate the idea.

**Weaknesses:**

1. The experiment is not sufficient, as discussed in Questions.

**Questions:**

1. My main concern is about the evaluation. Since the experiments are only conducted with 70 prompts, could you report the success rate of generation, i.e., how many generated results of the 70 prompts does not have Janus problem? This may be more convincing than the metrics used in the paper.

2. I have tried the score debiasing method in Stable-DreamFusion repo. However, based on my observation, the Janus problems are not alleviated. Is it true that the improvements is actually from Prompt Debiasing, and the improvement of Score Debiasing is quite marginal? If it is not true, could you provide an experiment solely on Score Debiasing and report the improvements of success rate of generation? (Success = Without Janus problem, Fail = With Janus problem.)

I am glad to raise my score if the concerns are resolved.

**Limitations:**

1. Since the results are based on NeRF with latent space rendering, the quality of generated 3D results is lower than current SOTA. Could this method be applied to other 3D representations, like DMTet, which demonstrates better quality than NeRF as is shown in Magic3D? Could this method be applied to a NeRF that renders in rgb space which demontrates stronger 3D consistency?

---

> ### Author Rebuttal · Authors · 2023-08-08
>
> **Success rate.** The reviewer raised a valid suggestion regarding the evaluation, especially concerning the success rate of generation. We concur that quantifying the number of generated results from the 70 prompts that do not exhibit the Janus problem would indeed provide compelling evidence. In light of this, as well as the user study in the supplementary material that evaluated the view consistency, faithfulness to user prompts, and overall quality, we have also conducted the evaluation the reviewer suggested, as presented below. The success rate applies to 41 out of 70 prompts, which feature countable faces, and we marked only those objects as successful that do not exhibit the Janus problem, i.e., those with the accurate number of faces. This experiment particularly corroborates our effectiveness in addressing the Janus problem, and we will include these results in a revised version of the paper.
>
> |               | Success-SJC | Success-Ours |
> |---------------|-------------|--------------|
> | **Success %** | 29.3%       | 68.3%        |
>
>
> **Effectiveness of score debiasing.** We acknowledge that the reviewer has experienced limited success in applying the score debiasing technique from the unofficial Stable-DreamFusion repository. It is worth noting that the effectiveness of score debiasing can vary depending on the baseline. As mentioned in our paper, if the 2D diffusion prior generates a severely deformed 3D field, our method might not fully address these issues. Nevertheless, to showcase the improvement achieved by applying score debiasing to the DreamFusion framework [Poole et al., 2023], we conducted an experiment only applying score debiasing (Fig. 2 in the general response). This experiment used the same method as proposed in DreamFusion and utilized DeepFloyd-IF, an RGB-pixel-based diffusion model similar to Imagen [Saharia et al, 2022]. It's important to note that, due to the scale differences between the RGB scores and latent scores, we adjusted our score debiasing threshold for RGB scores to one fourth of the value proposed in the paper. In the experiment with the high-performing framework, we see that the geometry issue decreases significantly when using only score debiasing.
>
> **Application on other 3D representations and NeRF in RGB space.** The method we have proposed, while tested with NeRF and latent space rendering, is fundamentally a technique applicable for a variety of text-to-3D generation frameworks using score distillation. As such, we believe it could indeed be applied to other 3D representations like DMTet or to a NeRF that renders in RGB space. To address this concern, in general response, we have provided a PDF file including results which demonstrate the applicability of our method to recent frameworks such as Magic3D (using DMTet) [Lin et al., 2023], DreamFusion (using NeRF in RGB space), and ProlificDreamer (using VSD) [Wang et al., 2023]. We kindly direct readers to the general response for more information.
>
> [Saharia et al., 2022] Photorealistic Text-to-Image Diffusion Models with Deep Language Understanding, NeurIPS 2022.
>
> [Poole et al., 2023] DreamFusion: Text-to-3D using 2D Diffusion, ICLR 2023.
>
> [Lin et al., 2023] Magic3D: High-Resolution Text-to-3D Content Creation, CVPR 2023.
>
> [Wang et al., 2023] ProlificDreamer: High-Fidelity and Diverse Text-to-3D Generation with Variational Score Distillation, arXiv preprint.

---

> > ### Comment · Reviewer_9gp3 · 2023-08-11
> >
> > I have read your rebuttal and it partially solves my concerns. I have raised my score to "weak accept".
> >
> > I suggest that you add the table of success rate and the experiments in the attached pdf into your paper somewhere in the main text or appendix, if accepted.

---

> > > ### Author Response · Authors · 2023-08-13
> > > **Response**
> > >
> > > Thank you for your feedback and for considering updating the scoring for the paper. I appreciate your suggestion and will revise our paper accordingly.

---

### Official Review · Reviewer_gQ7C · 2023-07-05

**Soundness:** 3 good
**Presentation:** 3 good
**Contribution:** 3 good
**Rating:** 6
**Confidence:** 3

**Summary:**

This paper addresses the Janus problem appearing in Score-Distillation-based 3D generation methods, where the most canonical view of an object appears in other views. In particular, the author propose two components for debiasing the score distillation and the prompt used for the generation. First, the authors provide a theoretical discussion over SDS loss and the terms that contribute to the Janus problem and bad generation quality. Therefore, the authors propose gradient clipping of the score gradient in order to tackle the discussed problem. Furthermore, the paper discusses another issue of the existing methods, which is the potential contradiction between tokens of the text prompt with the added view guidance tokens. To prevent the contradiction, the authors propose to exploit the point-wise mutual information (PMI) technique to identify and remove the contradictions from the text prompt. The proposed method evaluated quantitatively and qualitatively and compared to existing baselines. In addition, an ablation study on different proposed components is provided.

**Strengths:**

-- The paper is well-written.

-- The provided discussion on the bias of score distillation loss and view-augmented prompts is valuable

-- The novelty of the proposed components is sufficient.

-- Based on the provided results, the proposed method seems effective in alleviating the Janus problem.

**Weaknesses:**

-- The efficacy of the proposed method is still highly limited by the bias of diffusion models. (although this is an issue specific to the proposed method).

**Questions:**

-- I am wondering how much the proposed gradient clipping affects the convergence speed of the optimization. Is there any considerable difference with the baselines regarding the optimization steps?

**Limitations:**

Limitations and the societal impact of the work has been discussed in the supplementary material.

---

> ### Author Rebuttal · Authors · 2023-08-08
>
> **Bias of diffusion models.** This relates to the limitations of our proposed method including other approaches, specifically regarding the bias of diffusion models. As we mentioned in our paper, if the 2D prior generates a severely deformed 3D field in the baseline, our method might not be able to fully alleviate it. However, it should be noted that our method is intended for debiasing to better distill the 2D diffusion score, not for directly incorporating an additional 3D geometry prior. Nonetheless, our method can be used universally as diffusion models and score-distilling text-to-3D frameworks evolve in the future, acting as an effective plug-and-play method. Ambitiously, we hope that our work provides a stepping stone towards mitigating these biases and combining a 3D prior with those 2D score-based models in future research.
>
> **Convergence speed.** Regarding the question about the impact of gradient clipping on convergence speed, we assure the reviewer that the convergence speed is nearly unchanged by our approach (approximately 20 minutes for SJC and ours). Our experiments showed that the number of optimization steps is comparable to that of the baseline models. With the same number of steps, we observed that the content is maintained while being debiased, with fewer artifacts (figures in the main paper and the supplementary material are always optimized with $10,000$ steps). Additionally, in the general response, we display Fig. 4 in the PDF file to show what the rendered image at each step looks like as the scene undergoes optimization. The 3D scenes for both the baseline and ours evolve similarly in terms of optimization steps, with ours being debiased. Besides, this experiment also underscores the motivation for dynamic clipping of 2D-to-3D scores, as the geometry is determined in the early stages.
>
> **Applicability.** Our method is widely applicable to a variety of concurrent methods that use score distillation for text-to-3D models, including the very recent work on variational score distillation (VSD) in ProlificDreamer [Wang et al., 2023]. In general response, we have provided a PDF file including results which demonstrate the applicability of our method to recent frameworks such as Magic3D [Lin et al., 2023] (Fig. 3), DreamFusion [Poole et al., 2023] (Fig. 2), and ProlificDreamer (Fig. 1). We kindly direct the reviewer to the general response for more information.
>
> [Poole et al., 2023] DreamFusion: Text-to-3D using 2D Diffusion, ICLR 2023.
>
> [Lin et al., 2023] Magic3D: High-Resolution Text-to-3D Content Creation, CVPR 2023.
>
> [Wang et al., 2023] ProlificDreamer: High-Fidelity and Diverse Text-to-3D Generation with Variational Score Distillation, arXiv preprint.

---

> > ### Comment · Reviewer_gQ7C · 2023-08-11
> > **Final Comment**
> >
> > I thank the authors for their response and answering my question. I keep my initial score.

---

> > > ### Author Response · Authors · 2023-08-13
> > > **Response**
> > >
> > > We are grateful for your feedback and will further refine our paper. Thank you!

---

### Official Review · Reviewer_8eVk · 2023-07-06

**Soundness:** 3 good
**Presentation:** 3 good
**Contribution:** 3 good
**Rating:** 6
**Confidence:** 3

**Summary:**

This paper proposes debiased score sampling (D-SDS) for improving 2D diffusion-based 3D generation, targeting at the Janus problem. The method is composed of two parts: (i) score debiasing that cuts off scores from 2D diffusion and (ii) prompt debiasing that fixes the discrepancy between view prompts and object orientation. Experiments have been conducted from the baseline method SJC [Wang et al., 2023], where impressive improvements have been shown, especially in solving the Janus problem. A detailed ablation study has shown the value of each debiasing design.

[Wang et al., 2023] Score Jacobian Chaining: Lifting Pretrained 2D Diffusion Models for 3D Generation. In CVPR.

**Strengths:**

- This work is well-motivated, targeting a relevant problem of the Janus problem in 3D generation. The discussions on the problem and technical derivations are solid, insightful, and interesting. Besides, the paper is well-written.
- The proposed method is novel. I like the idea of debiasing scores and prompts, which makes good sense to me. Besides, the proposed method can be easily extended to other 2D diffusion-based methods.
- Experiments are well-designed and presented. Qualitative, quantitative, and user study experiments are shown, and the results demonstrate the effectiveness of solving the Janus problem.

**Weaknesses:**

- Somewhat not essential. The proposed method is simple and effective. However, it seems the method is only applicable to these SJC/DreamFusion 2D diffusion score-based methods. This is not a fundamental solution for the Janus problem but is kind of like a temporary effective trick that alleviates problems.
- Lacking complex examples. The presented results use relative prompts mainly for simple 3D objects. What about more complex 3D structures like the "Temple of Heaven" shown in SJC? Since one of the main difficulties of text-to-3D generation is for these more comprehensive geometries, it is important to show more of these cases. Is it another limitation of the proposed method?
- Limited baseline. The proposed method is currently conducted to improve the SJC baseline. What about other methods? Is it possible to be applied to the more recent work ProlificDreamer [Wang et al., 2023]?
- Minor suggestion: It seems that the Janus problem in 3D generation was first pointed out and termed for DreamFusion by Ben Poole before on [social media](https://twitter.com/poolio/status/1578045212236034048?s=20), it would be better to cite DreamFusion when first listing this issue in the paper. But it is okay if it is not cited for this term.

[Wang et al., 2023] ProlificDreamer: High-Fidelity and Diverse Text-to-3D Generation with Variational Score Distillation. arXiv preprint.

[Poole et al., 2023] DreamFusion: Text-to-3D using 2D Diffusion. In ICLR.

**Questions:**

I have no other questions besides those listed before. I am happy to increase the score if my concerns are addressed.

**Limitations:**

Since the method is based on SJC, it will strongly bound its performance.

---

> ### Author Rebuttal · Authors · 2023-08-08
>
> **Applicability.** Our method is widely applicable to a variety of concurrent methods that use score distillation for text-to-3D models, including the very recent work on variational score distillation (VSD) in ProlificDreamer [Wang et al., 2023]. In general response, we have provided a PDF file including results which demonstrate the applicability of our method to recent frameworks such as Magic3D [Lin et al., 2023] (Fig. 3), DreamFusion [Poole et al., 2023] (Fig. 2), and ProlificDreamer (Fig. 1). We kindly direct the reviewer to the general response for more information.
>
> **Complex examples.** The proposed method is designed to be also applicable to complex 3D structures, not just simple ones. In addition, it should be clarified that the "temple of heaven" example the reviewer suggested might not be the most suitable candidate for evaluating the Janus problem due to its radially symmetrical geometry. In fact, many papers showcase this type of prompts because the results are hardly identified as encountering the problem. Rather than this, we are eager to showcase our results with complex geometry that are particularly prone to issues like the Janus problem (e.g., "a majestic gryphon with the body of lion and the wings of an eagle'' or "an elegant teacup with delicate floral designs'') using the very recent implementation of threestudio [Guo et al., 2023], which enables us to generate more intricate objects with the ProlificDreamer VSD framework. To demonstrate our method's effectiveness in addressing the Janus problem in complex geometries, we present generation results in Fig. 1 in the general response and compare them with baseline models.
>
> **Baselines.** While we agree that applying our method to other models such as ProlificDreamer would further demonstrate its versatility, we selected SJC [Wang et al., 2023] as our primary baseline because it was the state-of-the-art text-to-3D framework available when we submitted our work. Actually, ProlificDreamer was published on arXiv after the NeurIPS paper submission deadline, and they even mentioned the debiasing methods in their appendix as orthogonal methods that can be used in conjunction with their variational score distillation approach.
>
> **Citation.** We appreciate the reviewer's suggestion to credit Ben Poole for introducing the Janus problem in social media. We will revise the manuscript to include the citation for DreamFusion.
>
> [Poole et al., 2023] DreamFusion: Text-to-3D using 2D Diffusion, ICLR 2023.
>
> [Wang et al., 2023] Score Jacobian Chaining: Lifting Pretrained 2D Diffusion Models for 3D Generation, CVPR 2023.
>
> [Lin et al., 2023] Magic3D: High-Resolution Text-to-3D Content Creation, CVPR 2023.
>
> [Wang et al., 2023] ProlificDreamer: High-Fidelity and Diverse Text-to-3D Generation with Variational Score Distillation, arXiv preprint.
>
> [Guo et al., 2023] threestudio: A unified framework for 3D content generation.

---

> > ### Comment · Reviewer_8eVk · 2023-08-12
> > **Post Rebuttal Comment**
> >
> > I highly appreciate the authors for the detailed response, which I suggest adding to the main paper appendix in the final version. My concerns are largely solved.
> >
> > Minor suggestion: It would be better to include some quantitative comparisons for methods like ProlificDreamer and Magic3D, if possible.

---

> > > ### Author Response · Authors · 2023-08-13
> > > **Response**
> > >
> > > We're pleased that your concerns have been largely resolved! We appreciate your comments and will revise our paper accordingly, hopefully including your minor suggestion.

---

### Author Rebuttal · Authors · 2023-08-08

**General response.** We deeply appreciate the insightful feedback provided on our manuscript and have thoroughly examined all comments. The reviewers recognized the critical nature of the Janus problem in text-to-3D generation, underscoring the importance of our contribution to the field. The reviewers acknowledged the strong motivation behind our research, emphasizing the relevance of the Janus problem within text-to-3D generation (8eVk, nz3q). They also praised the robustness of our discussions, technical insights, and the manuscript's lucidity (8eVk, gQ7C, 9gp3). Our innovative methodology has been spotlighted, especially the debiasing of scores and prompts approach, which has been recognized for its originality and potential adaptability to 2D diffusion-based methods (8eVk). In addition, our comprehensive experiments, covering qualitative, quantitative, and user studies, is a testament to our solution's efficacy in tackling the Janus problem (8eVk, gQ7C).

**Applicability.** Our method is designed to enhance 2D score-based text-to-3D generation methods. While it has been mainly claimed to be applicable to the SJC [Wang et al., 2023] and DreamFusion [Poole et al., 2023] frameworks, the applicability of our approach goes beyond these models. It can be adapted for any text-to-3D generation methods that leverage a score from a text-to-image diffusion model and use view-augmented prompting, which are the dominant approaches in current text-to-3D methods using the scores of 2D diffusion models [Chen et al., 2023; Poole et al., 2023; Wang et al., 2023; Lin et al., 2023; Seo et al., 2023]. These methods, including Magic3D [Lin et al., 2023] and ProlificDreamer [Wang et al., 2023], are to some extent susceptible to the Janus problem. We believe this context renders our work a rather novel solution that tackles the prevailing Janus problem. Thanks to the recent implementation of the text-to-3D frameworks [Guo et al., 2023], we have provided a PDF file including results that demonstrate the applicability of our method to recent frameworks such as Magic3D (Fig. 3), DreamFusion (Fig. 2), and ProlificDreamer (Fig. 1). We hope that concerns about the applicability are resolved when looking at the results in the file.

[Poole et al., 2023] DreamFusion: Text-to-3D using 2D Diffusion, ICLR 2023.

[Wang et al., 2023] Score Jacobian Chaining: Lifting Pretrained 2D Diffusion Models for 3D Generation, CVPR 2023.

[Lin et al., 2023] Magic3D: High-Resolution Text-to-3D Content Creation, CVPR 2023.

[Wang et al., 2023] ProlificDreamer: High-Fidelity and Diverse Text-to-3D Generation with Variational Score Distillation, arXiv preprint.

[Seo et al., 2023] Let 2D Diffusion Model Know 3D-Consistency for Robust Text-to-3D Generation, arXiv preprint.

[Chen et al., 2023] Fantasia3D: Disentangling Geometry and Appearance for High-quality Text-to-3D Content Creation, ICCV 2023.

[Guo et al., 2023] threestudio: A unified framework for 3D content generation.

---

### Decision · Program_Chairs · 2023-09-21

**Decision:**

Accept (poster)

**Comment:**

This paper was reviewed by four experts in the field. Based on the reviewers' feedback, the decision is to recommend the paper for acceptance to NeurIPS 2023. The reviewers did raise some valuable concerns that should be addressed in the final camera-ready version of the paper. The authors are encouraged to make the necessary changes to the best of their ability. We congratulate the authors on the acceptance of their paper!